# Pilot Study to Evaluate Serum Soluble Mesothelin-Related Peptide (SMRP) as Marker for Clinical Monitoring of Pleural Mesothelioma (PM): Correlation with Modified RECIST Score

**DOI:** 10.3390/diagnostics11112015

**Published:** 2021-10-29

**Authors:** Federica Grosso, Matilde Mannucci, Francesca Ugo, Paola Ferro, Maurizio Cassinari, Antonella Vigani, Antonina Maria De Angelis, Sara Delfanti, Michela Lia, Roberto Guaschino, Stefano Barbero, Silvio Roncella, Ugo Giannoni, Marinella Bertolotti, Maria Pia Pistillo, Vincenzo Fontana

**Affiliations:** 1Mesothelioma Unit, Azienda Ospedaliera SS Antonio e Biagio e Cesare Arrigo, 15121 Alessandria, Italy; antonina.deangelis@ospedale.al.it (A.M.D.A.); sara.delfanti@ospedale.al.it (S.D.); michela.lia@ospedale.al.it (M.L.); 2Translational Medicine, Dipartimento Attività Integrate Ricerca e Innovazione, Azienda Ospedaliera SS Antonio e Biagio e Cesare Arrigo, 15121 Alessandria, Italy; 3Clinical Epidemiology Unit, IRCCS Ospedale Policlinico San Martino, 16132 Genova, Italy; matilde.mannucci@hsanmartino.it (M.M.); vincenzo.fontana@hsanmartino.it (V.F.); 4Infrastruttura Ricerca Formazione e Innovazione, Dipartimento Attività Integrate Ricerca e Innovazione, Azienda Ospedaliera SS Antonio e Biagio e Cesare Arrigo, 15121 Alessandria, Italy; fugo@ospedale.al.it (F.U.); MBertolotti@ospedale.al.it (M.B.); 5Histopathology and Cytopathology Unit, Azienda Sanitaria Locale 5, 19121 La Spezia, Italy; paola.ferro@asl5.liguria.it (P.F.); roncy88@alice.it (S.R.); 6Laboratory Unit, Azienda Ospedaliera SS Antonio e Biagio e Cesare Arrigo, 15121 Alessandria, Italy; mcassinari@ospedale.al.it (M.C.); rguaschino@ospedale.al.it (R.G.); 7Oncology Division, Azienda Sanitaria Locale 5, 19121 La Spezia, Italy; antonella.vigani@asl5.liguria.it; 8Radiology Department, Azienda Ospedaliera SS Antonio e Biagio e Cesare Arrigo, 15121 Alessandria, Italy; stefano.barbero@ospedale.al.it; 9Radiodiagnostic Division, Azienda Sanitaria Locale 5, 19121 La Spezia, Italy; ugo.giannoni@asl5.liguria.it; 10Tumor Epigenetics Unit, IRCCS Ospedale Policlinico San Martino, 16132 Genova, Italy; mariapia.pistillo@gmail.com

**Keywords:** mesothelioma, mesothelin, serum biomarkers, modified RECIST, longitudinal study

## Abstract

A soluble mesothelin-related peptide (SMRP) is the only FDA-approved biomarker for diagnosis of pleural mesothelioma (PM) and the most used for monitoring treatment. Radiological assessment of PM, based on modified RECIST (mRECIST) criteria, is challenging. This pilot study was designed to evaluate whether SMRP levels correlated over time with mRECIST score. Serial serum samples from PM patients were collected and SMRP levels were measured and compared with the mRECIST score obtained through centralized CT scans by blinded review. The within-patient SMRP-mRECIST relationship over time was estimated through a normal random-effects regression approach applied to the log-transformed mRECIST score. Overall, 58 PM patients were included (46 males and 12 females) with a median age at diagnosis of 67 years (min–max = 48–79), 44 (76%) with epithelioid and 14 (24%) with non-epithelioid histology. The total number of SMRP measurements and CT scans considered for analysis was 183. There was a statistically significant correlation between SMRP and mRECIST score in the 2 cohorts considered both separately and jointly. These results, although exploratory, suggest that SMRP measurement might be considered as an adjunct to monitor PM patients in order to delay CT scans time interval, thus warranting further investigation.

## 1. Introduction

Pleural mesothelioma (PM) is a rare cancer of the mesothelial surfaces with dismal prognosis. PM patients have a median overall survival (OS) ranging from 4 to 13 months for untreated patients and from 6 to 18 months for treated patients [1,2]. Since 2004, platinum and antifolates have been the gold standard for the first line treatment of patients with advanced disease with a median OS of 12 months [3]. Recently, in the phase III pivotal trial CheckMate 743, the combination of nivolumab plus ipilimumab significantly prolonged OS with respect to standard chemotherapy [4]. Although there is no approved second line, the combination of ramucirumab and gemcitabine showed significant OS improvements over gemcitabine in the phase II RAMES study and nivolumab in the salvage setting remarkably improved progression-free and OS in the phase III placebo-controlled CONFIRM trial [5,6,7].

Despite the recent progress, there is still an urgent need for active therapies in this disease and for tools that could assist clinicians in improving the management of patients.

One of the most controversial issues, due to the unique morphology of tumor growth, is the evaluation of the radiological response to treatment not only in the context of clinical research but also in everyday practice. Specific response criteria, the so-called modified RECIST (mRECIST) criteria for PM, have been developed in 2004 and recently updated to help clinicians standardize the measurement of lesions thus allowing a better and more reproducible classification of response. The application of mRECIST criteria, however, requires expert and trained radiologists [8,9], and although the criteria are most often used in clinical trials, their use in clinical practice is still limited.

Serum biomarkers mirroring disease extension, in addition to the use of specific radiological criteria, would be of significant value and could represent an attractive option to optimize the clinical management of PM patients.

Among biomarkers, the most investigated is mesothelin, a membrane-bound glycoprotein, previously proposed for diagnostic use, produced at low levels by normal mesothelial cells and overexpressed in certain tumors including PM, pancreatic adenocarcinoma, ovarian, and lung cancer [10,11,12]. Serum soluble mesothelin-related peptide (SMRP) is the form of mesothelin released by PM cells into the blood where it can be detected and measured. Although subsequent evidence has indicated poor performance of SMRP as a screening test due to low sensitivity, it remains the most used and the only FDA-approved serum biomarker for diagnosis [13].

SMRP can be useful for the monitoring of therapy. Many studies have suggested a correlation between SMRP levels and PM response to systemic therapy [14,15,16,17]. A systematic review including 20 studies indicated that the decrease in serum levels of SMRP might correlate with response to treatment [18]. Moreover, in a small series of 10 PM patients followed at a single Italian institution, a longitudinal statistical analysis indicated that SMRP levels and pleural thickness showed very similar trends confirming a previously reported observation of a strong correlation between the two clinical markers [19].

This exploratory, descriptive pilot study was directed towards assessing the correlation between SMRP levels and disease extension, as described by the mRECIST criteria, during the patients’ clinical course. To this end, we used, for the first time, a statistical method able to estimate patient-specific rates of change in both clinical markers by analyzing the longitudinal profile of repeated measurements of the same markers.

## 2. Materials and Methods

### 2.1. Patients

This is a retrospective, observational pilot study designed to estimate the correlation over time between SMRP levels and mRECIST evaluations, independently of the anticancer treatment. The study population consisted of two different PM patients’ cohorts, the Alessandria (AL) cohort and the La Spezia (SP) cohort, including 40 and 18 patients, respectively. The AL cohort, followed by the Mesothelioma Unit at Azienda Ospedaliera SS Antonio e Biagio e Cesare Arrigo, Alessandria (Italy), was recruited from June 2014 to April 2020 and the follow-up period ended by April 2020. The SP cohort followed by the Oncology Unit at the Azienda Sanitaria Locale 5, La Spezia (Italy), was recruited from March 2010 to May 2015 and the follow-up period ended by September 2017. In both cohorts, the diagnosis was based on histological examination of tumor samples obtained by means of thoracoscopy or computed tomography (CT) guided biopsy and analyzed through the appropriate immunohistochemistry according to international reference guidelines [20].

In both cohorts, CT scans and peripheral blood examinations, inclusive of SMRP, were performed according to routine clinical practice. For each patient, we considered all CT scans performed along with the whole clinical history. In addition, we selected the SMRP measurement closest to the CT scans. The CT scans considered for analysis were stored in the AL and SP hospitals’ radiological archives. All the CT scans were retrieved, anonymized and sent for centralized review performed by the same expert radiologist, at the Azienda Ospedaliera SS Antonio e Biagio e Cesare Arrigo, Alessandria, Italy, blind to clinical data and SMRP levels. The disease extension was described through a numerical score measured according to the mRECIST criteria and referred to as the mRECIST score.

Patients’ characteristics, systemic treatments and CT scan reports, including the description of tumor extension and response to therapy, were collected in two separate databases at each participating center (AL and SP). A common database, including patients’ data from both cohorts, SMRP measurements and the mRECIST score over time was generated for the purpose of this study.

### 2.2. Measurement of Serum SMRP

Serum concentrations of SMRP were measured using the chemiluminescent enzyme immunoassay (CLEIA) kit (Lumipulse G Mesothelin, Fujirebio Europe, Ghent, Belgium) for the AL samples and the enzyme-linked immunosorbent assay (ELISA) kit (MesoMark, Fujirebio, Japan) for the SP samples, according to the respective manufacturers’ instructions.

Aliquots of each serum sample were centrifuged (1500× *g* for 10 min at 4 °C) and supernatants were stored at −20 °C until assays were performed. Each sample was tested in duplicate with deviation between duplicates lower than 10%. The lowest sensitivity thresholds were ≤0.1 nmol/L for the CLEIA kit and 0.3 nmol/L for the ELISA kit, respectively.

Results were expressed as the geometric mean of SMRP concentrations.

### 2.3. CT Imaging and Modified RECIST Score

All patients had CT scans according to routine practice. Patients who received chemotherapy had a CT scan every 3–4 months, whereas patients in the follow-up phase, or on treatment holidays, or candidates for best supportive care only, had a CT scan at clinically appropriate intervals (i.e., every 6–8 months) according to internal guidelines. In this framework, the pleural rind was measured through the sum of six perpendicular chest wall measurements from three separate sections of the pleural tumor, preferably above the carina. Tumor thickness perpendicular to the chest wall or mediastinum was measured in two positions at three separate levels on transverse CT slices; levels considered for measurements required to be at least 1 cm apart and related to anatomical landmarks to allow reproducible assessment at later time points. The sum of the six measurements defined a pleural unidimensional measure. Bidimensionally measurable lesions, such as mediastinal lymph nodes, were recorded unidimensionally, as for RECIST 1.1, and were added to the pleural measurement. Criteria for partial response and progressive disease were the same as in the original RECIST 1.1. system [9].

### 2.4. Statistical Analysis

Descriptive statistics were applied to explore patient and disease characteristics in the two cohorts separately and jointly. Categorical variables (gender, histology, ECOG-PS) were expressed as absolute and relative frequencies (percentages), whereas continuous variables (age at diagnosis, months from diagnosis and years of follow-up) were summarized using median value and inter-quartile range (IQR). The distributions of SMRP levels and mRECIST score were described using the geometric mean (GM) and corresponding 95% confidence limits (95% CL). Analysis of contingency tables and related chi-square test were performed to assess the correlation between categorical variables, whereas the Student’s *t*-test was applied to compare distributions of continuous variables among subgroups of patients.

The SMRP–mRECIST relationship over time was evaluated through a normal random-effects regression approach [21] applied to the log-transformed mRECIST (log-n-mRECIST) score. Such a statistical methodology is appropriate to manage two important data features: firstly, the random-effects regression allows consideration of the longitudinal structure of within-patient repeated measurements; secondly, the log-transformation is useful to fulfill the normality assumption, to reduce any undue influence on the regression results of aberrant data points and, at the same time, to estimate the geometric mean ratio (GMR), along with corresponding 95% CL, as an index of SMRP–mRECIST association over the follow-up period.

All analyses were performed using Stata (StataCorp. Stata Statistical Software. Release 16. College Station, TX, USA, 2020).

## 3. Results

### 3.1. Patients’ Characteristics

The study sample included 58 PM patients (40 from AL and 18 from SP) diagnosed in the period 2010–2017 and characterized by an individual profile composed of age at diagnosis, months from diagnosis to the first SMRP measurement, years of follow-up, gender, histology, Eastern Cooperative Oncology Group performance status (ECOG-PS), and vital status at the end of follow-up and treatment (Table 1). Each patient was followed from the date of first SMRP measurement until the date of death or last follow-up visit. Table 1 summarizes patients’ baseline characteristics, split in the two cohorts and analyzed together. In the whole population, median age was 67 years (IQR = 62–72), 46 (79.3%) were males and 12 (20.7%) females, histology was epithelioid in 44 (75.9%) patients and non-epithelioid in 14 (24.1%). The median time from diagnosis to the first determination of SMRP was 2.1 months (IQR = 0.9–12.2). Patients’ ECOG PS was ≤1 in 35 (60.3%) and >1 in 23 (37.7%). The median length of follow-up was 1.8 years (IQR = 1.0–2.7) and at the end of the study period only six patients (10.3%) were still alive.

Regarding treatment, in the AL cohort 6 (15.0%) patients received standard chemotherapy, 13 (32.5%) chemotherapy and immunotherapy, 21 (52.5%) chemotherapy and antiangiogenics. In the SP cohort 15 (83.3%) patients received chemotherapy and 3 (16.6%) best supportive care only.

At baseline, some differences between the statistical profiles of the two cohorts were highlighted (Table 1). AL patients had better daily living performance (ECOG-PS ≤ 1: 70.0% vs. 38.9%) and the SP cohort showed a median follow-up period which was approximately twice (2.5 years; IQR = 1.7–3.1) that observed in the AL cohort (1.4 years; IQR = 0.9–2.4). In addition, a death risk excess of approximately 25% was found in the AL patients (rate = 41.8 per 1000 per month; 95% CL = 29.8–58.4) when compared to the SP patients (rate = 33.8 per 1000 per month; 95% CL = 21.3–53.6) (Table 1).

### 3.2. SMRP and mRECIST Measurements

Over the follow-up period, AL and SP patients had 141 and 42 mRECIST measurements, respectively, corresponding to an annual mean frequency of 2.1 and 1.0 per patient. Consistently, these measurements were selected for the analysis using the time closeness to each CT scan examination as a selection criterion for the SMRP measurements. Using this framework, the median time between each CT scan examination and the corresponding SMRP measurement was 7 days (IQR = 4–13) for AL patients and 30 days (IQR = 16–50) for SP patients.

Table 2 shows the distributions of SMRP levels and mRECIST score values at baseline by cohort and histological subtype. In both cohorts, the GM of SMRP levels was 2.77 nM (95% CL = 2.11–3.66) in epithelioid patients and 1.89 nM (95% CL = 1.16–3.09) in non-epithelioid patients. GM levels of SMRP were slightly lower in AL patients (2.42 nM, 95% CL = 1.74–3.36 vs. 2.81 nM, 95% CL = 2.03–3.88); the imbalance was mainly due to differences between non-epithelioid PM patients (1.43 nM, 95% CL = 0.76–7.70 vs. 3.81 nM, 95% CL = 1.92–7.56). Similar results were obtained for the mRECIST distribution. AL patients scored on average lower (GM = 82.5, 95% CL = 69.5–97 vs. GM = 94.4, 95% CL = 77.0–115.6), although in this case much of the observed divergence was observed within the epithelioid PM subgroup (GM = 81.8, 95% CL = 67.0–99.8 vs. GM = 97.1, 95% CL = 76.6–123.0).

### 3.3. Relationship between SMPR and mRECIST

The results of the random-effects regression analysis are shown in Table 3. The SMRP-mRECIST association was evaluated in two different ways. Firstly, SMRP entered the regression equation as a four-category variable derived by dividing the biomarker levels according to the quartiles of the original measurements. Secondly, SMPR was used as a base-2-log-transformed continuous variable (log-2-SMRP). In both cases, a clear upward trend in mRECIST score was estimated as a function of increasing SMRP levels. Specifically, using the first SMRP category as a reference, mRECIST GM values showed a positive change of approximately 15–20% on average as SMRP moved from one category to the next highest. This result was consistent with that obtained by replacing the categorical terms of SMRP with log-2-SMRP. In this case, it was observed that when the SMRP levels doubled (i.e., increased by 100%) the proportional change in mRECIST score was approximately +16% in the AL cohort (GMR = 1.16, 95% CL = 1.11–1.22; *p*-value < 0.001) and +18% in the SP cohort (GMR = 1.18, 95% CL = 1.06–1.31; *p*-value < 0.003) and, as a consequence, a similar result was obtained for the joint-cohort analysis (GMR = 1.19, 95% CL = 1.13–1.25; *p*-value < 0.001). It is worth noting that overlapping results were also computed for the two histological subtypes (epithelioid: GMR = 1.20, 95% CL = 1.13–1.27; non-epithelioid: GMR = 1.17, 95% CL = 1.10–1.25).

Figure 1 depicts the estimated association in terms of trajectories in the log-n-mRECIST score over log-2-SMRP levels in each patient (thin lines) and averaged across all patients (thick lines) in both cohorts separately and jointly.

Figure 2 shows two examples of consistency between SMRP level decrease (Figure 2A) or increase (Figure 2B) and mRECIST measurements/disease extension (Figure 2A: disease treatment response; Figure 2B: disease progression). These examples are distinguished by their subheadings. They provide a concise and precise description of the experimental results, their interpretation, as well as the experimental conclusions that can be drawn.

## 4. Discussion

This exploratory pilot study was performed to assess the correlation between time changes in SMRP levels and in disease extension, as described by the mRECIST scores, in a sample of 58 PM patients followed at two hospitals in the north of Italy, both located in areas (Alessandria and La Spezia Provinces) with a high incidence of PM. It was undertaken to confirm a previous observation, made in a smaller series of PM patients, that SMRP levels and pleural thickness had very similar trends [19]. On this occasion, we chose the mRECIST score instead of the pleural thickness because it was the standardized way to describe disease extension and represented the basis of the criteria for response evaluation. We decided to analyze two different cohorts of patients together to increase the number of subjects and therefore the statistical power, but also to evaluate the association in two different circumstances.

In our study, the within-patient correlation over time between SMRP level and mRECIST changes was evaluated independently of systemic treatment or follow-up phase.

To determine the correlation, we used a normal random-effects regression approach [21] applied to the log-transformed mRECIST. Such a statistical tool has the clear advantage over other regression methods of indicating the joint time trajectory of both clinical markers within each patient. In this way, positive or negative changes in both markers can be mainly attributed to the corresponding positive or negative changes in disease extension given that all comparisons were based on longitudinal repeated measurements undertaken on the same patient. The final result was represented by a single index of association (GMR) derived as an average computed over all patients which summarizes the joint marker behavior during the whole follow-up period. To our knowledge, such a statistical approach has not yet been applied in a PM clinical setting to evaluate the direction and intensity of association between the two markers.

Our results showed a statistically significant correlation between SMRP and mRECIST score in the two cohorts when analyzed both separately and jointly. An mRECIST score mean increase of approximately 20% was estimated for each doubling in SMRP levels when the data were considered together. Similar results were also obtained for the two histology sub-types (epithelioid: +20%; non-epithelioid: +17%).

Although the two cohorts differed in some baseline characteristics, in particular oncological treatments, length of follow-up and method used to measure SMRP levels (CLEIA in the AL cohort and ELISA in the SP cohort), all the comparisons were performed within patients, namely by taking into account all patient-specific features, thus providing a more accurate estimation of the SMRP-mRECIST relationship over time. The small sample size and the lack of a specific samples collection timeline do though represent a weakness of the present study, which was aimed at producing a proof of concept for the use of SMRP as a marker for the clinical monitoring of PM.

Blood biomarkers whose levels reflect disease extension are very useful in the clinical management of cancer patients and the availability of a reliable biomarker for PM patients is of great importance, since these patients are currently monitored only with radiological examinations that are often very difficult to interpret. SMRP has been demonstrated to be a useful biomarker for the monitoring of treatment. In a surgical series of 102 patients, SMRP level decreased immediately after surgery and increased over time during disease progression in 82.4% of patients with progressive disease suggesting that it is a promising serum biomarker for the detection of recurrence after resection that may have value in clinical practice [14]. In a series of 40 patients with SMRP > 1 nM and receiving chemotherapy and immunotherapy, an increase in serial measurements of 10% or greater correlated with a probability of 75% of radiological progression using mRECIST criteria and with worse OS [22]. In a series of 41 PM patients receiving chemotherapy or best supportive care, a 10% rise in SMRP could predict radiological progression with a sensitivity of 96% and a specificity of 74% [23]. Other studies have confirmed the value of SMRP for the monitoring of therapy; a systematic review including 20 studies (18 prospective and 2 retrospective) for a total of 1578 patients suggested that SMRP measured before and after treatment can track treatment response as seen on serial CT scans [18].

A blood biomarker that could have a role in addition to radiological assessment reflecting response to treatment would be of great utility for clinicians managing PM.

Due to the peculiar pattern of growth of PM, disease evaluation is quite difficult and remains an open challenge with a higher disagreement between radiologists than for other solid tumors. Modified RECIST criteria, defined since 2004, have enabled more accurate measurements of the pleural rind by implementing a standardized way to measure the tumor burden. Although this has brought an improvement in disease evaluation, high variability and imprecision in measurements remain and the application of mRECIST criteria still requires very expert and trained radiologists. As an alternative, tumor volume measurement on cross-sectional imaging has emerged as a potential tool with prognostic significance in PM [24,25]. In the last decade, this strategy has been evaluated as a potentially more reliable measure of chemotherapy response and predictor of outcomes in PM than RECIST [26]. Although promising, further validation of the role of tumor volume in clinical staging is required with large international studies before it can be incorporated into the clinical staging algorithm [27].

In our study to ensure the consistency in the evaluation of the mRECIST score we required all the CT scans to be reviewed by an expert radiologist blind to the clinical data and to the SMRP levels. Our results suggest that a doubling in SMRP level corresponded to an increase in mRECIST score of 16% and 18% in the AL and SP cohorts, respectively, and of approximately 20% when the two cohorts were analyzed together. According to mRECIST criteria, an increase of the mRECIST score of 20% is consistent with progressive disease. This finding, if confirmed in larger series, might provide new insight into the correlation between SMRP increase and disease extension changes.

The possibility of monitoring disease progression in PM patients is becoming increasingly important given the recent advances in systemic therapy, the promising results from second-line chemotherapy trials, as well as the availability of second-line experimental protocols.

One of the most controversial points among studies is the variation in the thresholds used to define a significant change in serum SMRP, as well as the appropriate sampling intervals during or after treatment. Our study suggests that the doubling of SMRP approximates a radiological progression disease based on mRECIST criteria. Guidelines on follow-up generically suggest that a 3/4-month interval is good practice, without indicating which kind of examinations should be performed. SMRP determinations could be readily used in the monitoring of patients in conjunction with, or even as an alternative to, radiological exams.

## 5. Conclusions

In conclusion, our exploratory proof of concept study has confirmed that SMRP could be a reliable biomarker for use in the monitoring of PM progression and suggests that the biomarker might assist in more accurate disease management and more efficient clinical decision-making. However, these data must be taken with caution and warrant further confirmatory cohort studies.

## Figures and Tables

**Figure 1 diagnostics-11-02015-f001:**
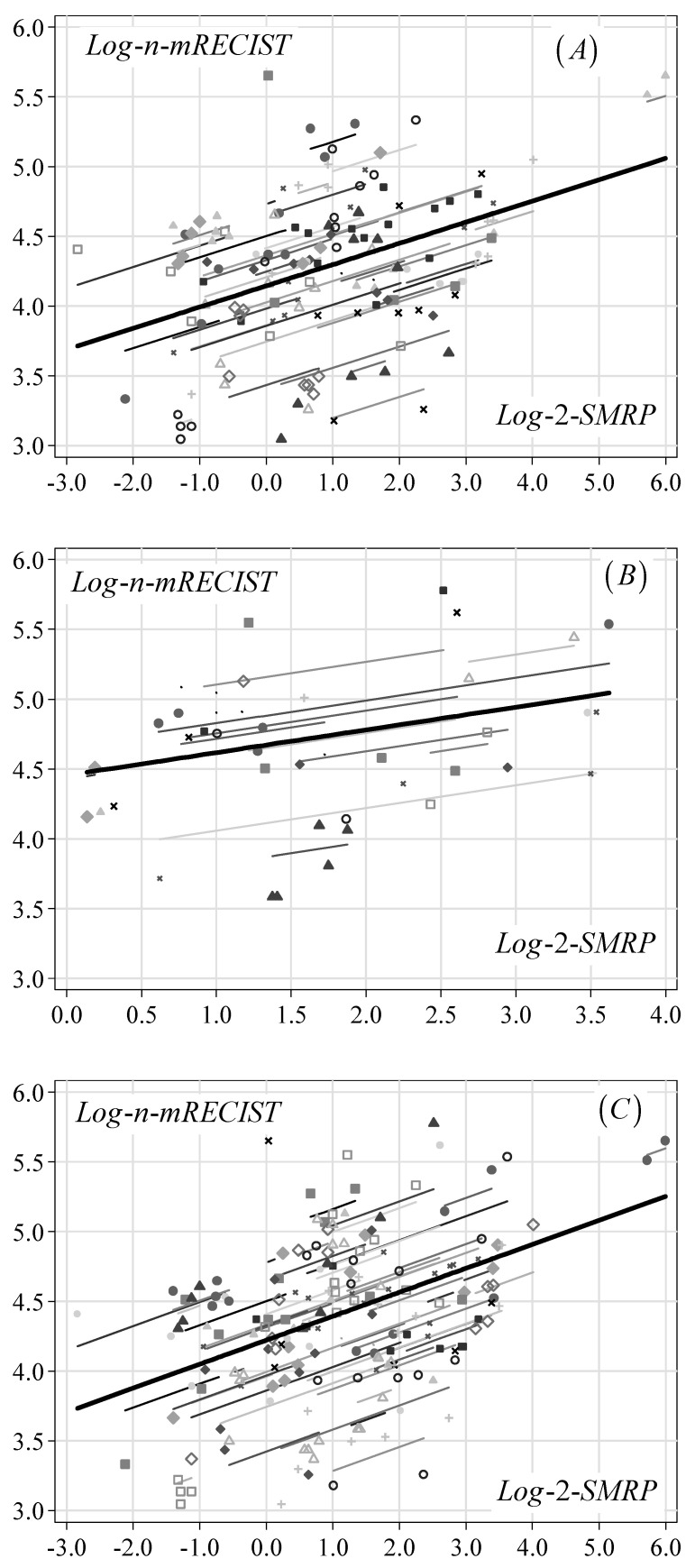
Association between mRECIST score and SMRP levels (double logarithmic scale) estimated through a normal random-effects regression in the AL cohort (**A**), SP cohort (**B**) and both cohorts (**C**). Note: mRECIST: modified RECIST; SMRP: soluble mesothelin-related peptides; log-n: natural logarithm; log-2: base-2 logarithm. Symbols: observed patient-specific levels of log-n-mRECIST score and log-2-SMRP. Thin lines: patient-specific linear relationships between log-n-mRECIST score and log-2-SMRP levels. Thick lines: overall relationship obtained as an average across all patients.

**Figure 2 diagnostics-11-02015-f002:**
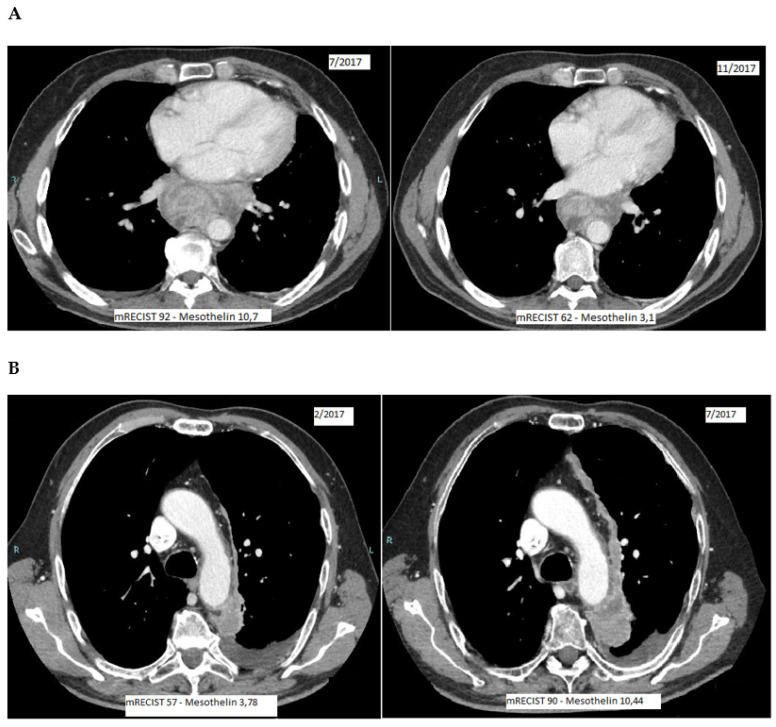
Examples of correlation between mRECIST score and SMRP levels. (**A**) mRECIST reduction (disease treatment response) correlates with SMRP reduction; (**B**) mRECIST increase (disease progression) correlates with SMRP increase; mRECIST: modified RECIST; SMRP: soluble mesothelin-related peptides.

**Table 1 diagnostics-11-02015-t001:** Patients’ characteristics in MPM cohorts.

Variable	AL Cohort				SP Cohort				Both Cohorts			
	No.	%	P50	IQR	No.	%	P50	IQR	No.	%	P50	IQR
Age at diagnosis			67	61–72			68	63–72			67	62–72
Days from diagnosis			5	3–27			3	0–56			5	2–30
Years of follow-up			1.4	0.9–2.4			2.5	1.7–3.1			1.8	1.0–2.7
Mortality rate/1000/month ^1^			41.8	29.8–58.4			33.8	21.3–53.6			38.6	29.4–50.7
Gender												
Male	29	72.5			17	94.4			46	79.3		
Female	11	27.8			1	5.6			12	20.7		
Histology												
Epithelioid	30	66.7			14	77.8			44	75.9		
Non-epithelioid	10	33.3			4	22.2			14	24.1		
ECOG-PS												
≤1	28	70.0			7	38.9			35	60.3		
>1	12	30.0			11	61.1			23	39.7		
Vital status												
Alive	6	15.0			0	0.0			6	10.3		
Dead	34	85.0			18	100.0			52	89.7		
Therapy												
Chemotherapy	6	15.0			15	83.3			21	36.2		
Chemo and immunotherapy	13	32.5			0	0.0			13	22.4		
Chemo and antiangiogenic	21	52.5			0	0.0			21	36.2		
Supportive care	0	0.0			3	16.6			3	5.2		
Whole sample	40	100.0	-	-	18	100.0	-	-	58	100.0	-	-

AL: Alessandria; SP: La Spezia; P50: median value; IQR: inter-quartile range. ^1^ Mean rate and corresponding 95% confidence interval.

**Table 2 diagnostics-11-02015-t002:** Determinations of SMRP levels and mRECIST score in all MPM patients and by histological subtypes.

Clinical Marker	Histology	AL Cohort		SP Cohort		Both Cohorts	
		GM	95% CL	GM	95% CL	GM	95% CL
SMRP levels (nM)	Epithelioid	2.88	1.99–4.16	2.57	1.78–3.71	2.77	2.11–3.66
	Non-epithelioid	1.43	0.76–7.70	3.81	1.92–7.56	1.89	1.16–3.09
	Whole sample	2.42	1.74–3.36	2.81	2.03–3.88	2.53	1.98–3.23
mRECIST score	Epithelioid	81.8	67.0–99.8	97.1	76.6–123.0	86.3	74.2–100.5
	Non-epithelioid	84.7	59.9–119.7	85.5	55.0–133.1	84.9	64.9–111.2
	Whole sample	82.5	69.5–97.8	94.4	77.0–115.6	86.0	75.4–98.0

SMRP: soluble mesothelin peptide; mRECIST: modified RECIST; GM: geometric mean; 95% CL: 95% confidence limits for GM.

**Table 3 diagnostics-11-02015-t003:** Association between SMRP levels and mRECIST score estimated through a log-normal random-effects regression method in MPM cohorts.

AL Cohort					SP Cohort					Both Cohorts				
SMRP (nM)	GM	GMR	95%-CL	*p*-Value	SMRP (nM)	GM	GMR	95%-CL	*p*-Value	SMRP (nM)	GM	GMR	95%-CL	*p*-Value
Categories (median)				0.010	Categories (median)				0.043	Categories (median)				0.006
0.14–0.99 (0.53)	61.1	1.00	(Ref.)		1.10–1.99 (1.61)	95.3	1.00	(Ref.)		0.14–1.14 (0.61)	73.3	1.00	(Ref.)	
1.00–1.71 (1.39)	68.8	1.13	0.97–1.31		2.00–2.78 (2.42)	104.4	1.09	0.83–1.44		1.15–2.01 (1.57)	77.2	1.05	0.90–1.24	
1.72–3.57 (2.55)	77.2	1.26	1.06–1.51		2.79–5.72 (3.65)	123.6	1.29	1.00–1.69		2.02–3.94 (2.79)	90.2	1.23	1.02–1.49	
3.58–63.9 (7.15)	84.4	1.38	1.14–1.67		5.73–12.3 (9.10)	134.5	1.41	1.06–1.88		3.95–63.9 (7.15)	99.7	1.36	1.13–1.64	
Log-2-levels				<0.001	Log-2-levels				0.003	Log-2-levels				<0.001
Linear trend		1.16	1.11–1.22		Linear trend		1.18	1.06–1.31		Linear trend		1.19	1.13–1.25	

SMRP: soluble mesothelin peptide; mRECIST: modified RECIST; GM: mRECIST geometric mean; GMR: mRECISTgeometric mean ratio, i.e., ratio between the mRECIST GM in each SMRP category and mRECIST GM in the first SMRP category; Ref.: SMRP reference category; median: SMRP median value in each SMRP category; 95%-CL: 95% confidence limits for GMR; *p*-value: probability level associated with the likelihood ratio test; Log-2-levels: base-2 log-transformedSMRP levels.

## Data Availability

Patients’ dataset (both cohorts) available.

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
