# Peer review of "Pilot Study to Evaluate Serum Soluble Mesothelin-Related Peptide (SMRP) as Marker for Clinical Monitoring of Pleural Mesothelioma (PM): Correlation with Modified RECIST Score"

_diagnostics, 2021, doi:10.3390/diagnostics11112015_

Round 1

Reviewer 1 Report

Which method and how should the ideal response measurement be made in mesothelioma? What are the response or progression criteria? Answers to these and similar questions are still being sought. There is still no accepted method other than radiology. Therefore, this study is important. However, the study is methodologically complex and does not guide the clinician. Analyzing all the results together is confusing. How did the radiological and mesothelin measurements progress corresponding to the time points at which treatment response was evaluated? It has been stated that doubling the SMRP level corresponds to an increase of approximately 20% radiologically, so it can be interpreted as progression. However, it is unclear how much the SMRP level decreased in responding patients. I think it would be appropriate to define the statistical method better and to express the results accordingly.

Author Response

Reviewer 1

Comments and Suggestions for Authors

Which method and how should the ideal response measurement be made in mesothelioma? What are the response or progression criteria? Answers to these and similar questions are still being sought. There is still no accepted method other than radiology. Therefore, this study is important. However, the study is methodologically complex and does not guide the clinician.

  1. Analyzing all the results together is confusing.
    - We think that joint data analysis and related results were necessary because the aim of our investigation was to give evidence of a common time tendency of both clinical markers in both cohorts regardless of the disease status of each patient, however stable, progressive or responsive they could be.

  2. How did the radiological and mesothelin measurements progress corresponding to the time points at which treatment response was evaluated?
    - As above mentioned, the goal of our analysis was not to evaluate the changes of SMRP caused by treatment effect, but to estimate the modification of mRECIST with respect to the changes of SMRP.

  3. It has been stated that doubling the SMRP level corresponds to an increase of approximately 20% radiologically, so it can be interpreted as progression. However, it is unclear how much the SMRP level decreased in responding patients. I think it would be appropriate to define the statistical method better and to express the results accordingly.
    - The statistical method used (random effects regression) is not a novel but a well-established methodology clearly cited in the text with the reference included in the bibliography as per journal guidelines. In addition, data manipulations (categorization and log-transformation) were applied not only to fulfil the statistical modelling assumptions but also to provide readers, above clinical researchers, with easily interpretable results. In this respect, we allow ourselves to say that the expression “a mRECIST score mean increase of about 20% was estimated for each doubling in SMRP levels”, quoted by the Reviewer, is a clear take-home message which derives directly from the statistical analysis.

Reviewer 2 Report

A prospective study using soluble mesothilin related peptides(SMRP) for pleural mesothelioma (PM) patients from 2 clinical trials found that SMRP and modified RECIEST score are good prognostic tools. Though the findings are positive and interesting, there are some issues for the manuscript.

  1. There is no new finding since SMRP and RECIEST have been studied on previous papers.
  2. The patient groups are heterogenous from 2 trials. The patients population, treatment protocol and even the test method of SMRP are different. 
  3. Patient number is small for the validate the findings, especially with heterogenous populations.

Author Response

Reviewer 2

Comments and Suggestions for Authors

A prospective study using soluble mesothelin related peptides (SMRP) for pleural mesothelioma (PM) patients from 2 clinical trials found that SMRP and modified RECIEST score are good prognostic tools. Though the findings are positive and interesting, there are some issues for the manuscript.

  1. There is no new finding since SMRP and RECIST have been studied on previous papers. This study is intended to strengthen the observation already made in a previous paper on smaller case series, as reported in the introduction.
    The patient groups are heterogeneous from 2 trials. The patient population, treatment protocol and even the test method of SMRP are different.
    - As stated in the text, this is an observational study, not a clinical trial and the data of two groups do not derive from two independent trials. Accordingly, a remarkable level of heterogeneity between the two cohorts and among pts within each cohort is definitely present. We would like to point out that the statistical method chosen allowed us to minimize such a drawback firstly by taking into consideration the within-pts longitudinal changes in both clinical markers (through the random effects component) and secondly by adjusting for imbalances in the individual characteristics of each pts (through the regression component). The method used to measure serum concentration of SMRP is described in the appropriate paragraph. The different method used in the two cohorts has no impact on the final message of the study, since the goal was to describe the changes of SMRP with respect to the changes of mRECIST.
  1. Patient number is small to validate the findings, especially with heterogeneous populations.
    - We acknowledge that the sample size is relatively small, even though pleural mesothelioma is a very rare tumor. But we would like to highlight that while heterogeneity is involved in the clinical validity of our investigation (see point 1), sample size has to do with the precision of the indexes of association, in this case the GMR. As it can be seen from Table 3, both in the categorical and log-transformed analysis, regardless of the statistical significance (p-value), all GMR estimates show a high precision based on very narrow 95%-CLs.

Reviewer 3 Report

Gross and colleagues investigate the association between changes in SMRP and mRECIST measurements of disease progression. I have the following specific comments:

  • The basis of this analysis is fundamentally flawed. If mRECIST is a poor indicator of disease status, why is a biomarker (SMRP) being correlated with mRECIST?
  • This is labelled a prospective descriptive study but it sounds like all the data were gathered retrospectively. If it was in fact prospective, please clarify how patients were selected for consent and what IRB reviewed the protocol to ensure compliance with appropriate standards for clinical research. If it was actually retrospective, please correct the relevant descriptors.
  • Page 3, line 11, states CT scan performed by the patients… probably should be “of the patients”.
  • What about the temporal changes in SMRP related to treatment and cell death? How is this accounted for? With such a small sample size, the lack of standardization (which would be present if this were truly a prospective trial) in blood draw timing relative to treatment and imaging can have substantial influence on the results and should be addressed as a weakness/limitation of the work.
  • The discussion is lacking in the consideration of novel imaging measurement systems such as volumetric CT.
  • The time interval of 3-4 months for people on active treatment is not consistent with US standards of care and should be discussed as a limitation of this work.
  • The conclusion is not supported by the data. This is a small and inconsistent study *suggesting* that SMRP may be a reliable marker for disease status in mesothelioma. To assert that this study is confirmatory of SMRP as a reliable biomarker is very much overstated and does not consider the limitations of size, geography, practice patterns, and inconsistent timing of the blood draws relative to treatment and imaging.

Author Response

Reviewer 3

Comments and Suggestions for Authors

Grosso and colleagues investigate the association between changes in SMRP and mRECIST measurements of disease progression. I have the following specific comments.

  1. The basis of this analysis is fundamentally flawed. If mRECIST is a poor indicator of disease status, why is a biomarker (SMRP) being correlated with mRECIST?
    -Despite their limitations (which are mostly related to the radiologist experience and expertise), mRECIST criteria are the gold standard for mesothelioma evaluation both in the clinical trials and in the daily clinical practice.

  2. This is labelled a prospective descriptive study but it sounds like all the data were gathered retrospectively. If it was in fact prospective, please clarify how patients were selected for consent and what IRB reviewed the protocol to ensure compliance with appropriate standards for clinical research. If it was actually retrospective, please correct the relevant descriptors.
    - Actually, the correct definition should be ‘non-concurrent prospective’ study. In practice, in this type of study researchers went back in time to the first available clinical assessment (sometimes at diagnosis, sometimes later) of each pt enrolled and then retrieved the significant information from clinical records and notes up to the present time. Data are therefore merged and analyzed. Following the comment of the reviewer we amended the manuscript accordingly. Also, as stated in the paper, the study was approved by the Ethics Committee of Azienda Ospedaliera SS Antonio e Biagio e Cesare Arrigo, Alessandria (authorization 1704; 27/10/2020) for the AL cohort and by the Ethics Committee of the Liguria Region (P.R. 207REG2014) for the SP cohort. Informed consent was obtained from all subjects involved in the study.

  3. Page 3, line 11, states CT scan performed by the patients… probably should be “of the patients”.
    - We modified the text accordingly.

  4. What about the temporal changes in SMRP related to treatment and cell death? How is this accounted for? With such a small sample size, the lack of standardization (which would be present if this were truly a prospective trial) in blood draw timing relative to treatment and imaging can have substantial influence on the results and should be addressed as a weakness/limitation of the work.
    -We amended the text according to this comment.

  5. The discussion is lacking in the consideration of novel imaging measurement systems such as volumetric CT. The time interval of 3-4 months for people on active treatment is not consistent with US standards of care and should be discussed as a limitation of this work.
    - The radiological assessments have been performed according to local clinical practice guidelines. We added in the discussion the consideration of novel imaging measurement systems such as volumetric CT, as per Reviewer’s comment.

  6. The conclusion is not supported by the data. This is a small and inconsistent study *suggesting* that SMRP may be a reliable marker for disease status in mesothelioma. To assert that this study is confirmatory of SMRP as a reliable biomarker is very much overstated and does not consider the limitations of size, geography, practice patterns, and inconsistent timing of the blood draws relative to treatment and imaging.
    - We amended the conclusions according to the Reviewer’s comment.

Round 2

Reviewer 1 Report

None

Reviewer 2 Report

Though the authors modified the manuscript according to previous suggestion, the small number of patients and not much new information from this retrospective study make it not suitable for publication in the journal.